Factors affecting livestock depredation by snow leopards (Panthera uncia) in the Himalayan region of Nepal

Karki Ajay clickajaya@gmail.com 1
Panthi Saroj 2
1 Ministry of Forests and Environment , Kathmandu , Nepal
2 Ministry of Industry Tourism Forest and Environment, Pokhara , Gandaki , Nepal
Yoccoz Nigel
Electronic publication date: 2021 Jun 16
Publication date: 2021
Volume: 9
Electronic Location ID: e11575
Received 2020 Aug 14; Accepted 2021 May 18
Copyright: ©2021 Karki et al.
Copyright year: 2021
Copyright holder: Karki et al.
License: This is an open access article distributed under the terms of the Creative Commons Attribution License, which permits unrestricted use, distribution, reproduction and adaptation in any medium and for any purpose provided that it is properly attributed. For attribution, the original author(s), title, publication source (PeerJ) and either DOI or URL of the article must be cited.
License URL: https://creativecommons.org/licenses/by/4.0/

Keywords: Conflict, Habitat, Himalaya, Livestock depredation, Modeling, Snow leopard, Wildlife management

Funding: Government of Nepal, Department of Forest The Government of Nepal, Department of Forest allocated the funding to support this study. The funders had no role in study design, data collection and analysis, decision to publish, or preparation of the manuscript.

==============================
The snow leopard (Panthera uncia) found in central Asia is classified as vulnerable species by the International Union for Conservation of Nature (IUCN). Every year, large number of livestock are killed by snow leopards in Nepal, leading to economic loss to local communities and making human-snow leopard conflict a major threat to snow leopard conservation. We conducted formal and informal stakeholder’s interviews to gather information related to livestock depredation with the aim to map the attack sites by the snow leopard. These sites were further validated by district forest office staffs to assess sources of bias. Attack sites older than 3 years were removed from the survey. We found 109 attack sites and visited all the sites for geo location purpose (GPS points of all unique sites were taken). We maintained at least a 100 m distance between attack locations to ensure that each attack location was unique, which resulted in 86 unique locations. A total of 235 km2 was used to define livestock depredation risk zone during this study. Using Maximum Entropy (MaxEnt) modeling, we found that distance to livestock sheds, distance to paths, aspect, and distance to roads were major contributing factors to the snow leopard’s attacks. We identified 13.64 km2 as risk zone for livestock depredation from snow leopards in the study area. Furthermore, snow leopards preferred to attack livestock near livestock shelters, far from human paths and at moderate distance from motor roads. These identified attack zones should be managed both for snow leopard conservation and livestock protection in order to balance human livelihoods while protecting snow leopards and their habitats.

Introduction

The snow leopard (Panthera uncia) is a wild carnivore native to 12 countries in central Asia (China, Bhutan, Nepal, India, Pakistan, Afghanistan, Tajikistan, Uzbekistan, Kyrgyzstan, Kazakhstan, Russia, and Mongolia) (McCarthy et al., 2017). The home range of this species is 124–207 km2 (Johansson et al., 2016) but estimated at 11–37 km2 in Nepal’s Himalaya (Jackson, 1996). In Qilianshan National Nature Reserve, China, the density of snow leopard is 3.31 individuals per 100 km2 (Alexander et al., 2015). Nepal has extremely varying population density; for example Langu valley has 10–12 animals per 100 km2 and Manang has 5–7 animals per 100 km2 (DNPWC, 2017). The primary prey targeted by snow leopards include wild species such as blue sheep (Pseudois nayaur) and marmots (Marmota caudate) as well as domesticated livestock such as yak (Bos grunniens) and sheep (Ovis spp.) (Aryal et al., 2014; Weiskopf, Kachel & McCarthy, 2016). Snow leopards co-exist with other Himalayan carnivores, such as red fox (Vulpes vulpes), grey wolf (Canis lupus), Eurasian lynx (Lynx lynx) and dhole (Cuonal pinus) (Alexander et al., 2016a; Bocci et al., 2017). Male snow leopards represent a greater threat to livestock than females (Chetri, Odden & Wegge, 2017). While there are several studies characterizing snow leopards, their habits and habitats, there is a need for more localized information to improve conservation management practices.

Human-snow leopard conflict, especially related to livestock depredation, represents a major threat to snow leopards (Li et al., 2013; Mijiddorj, Alexander & Samelius, 2018; Suryawanshi et al., 2013; Ud Din et al., 2017; Wegge, Shrestha & Flagstad, 2012). Livestock grazing in snow leopard habitat has been seen to be a serious conservation threat to this species (Ghoshal et al., 2017; Khanal et al., 2018; Sharma, Bhatnagar & Mishra, 2015). One of the main stressors of snow leopard poaching was found to be retaliatory killing as a consequence of livestock depredation (Maheshwari & Niraj, 2018). Another important factor influencing snow leopard poaching is the illegal trade of the body parts and pelts (Hussain et al., 2003), which is also on the rise (Li & Lu, 2014). Furthermore, impacts of climate change have emerged as a primary threat to snow leopards; their habitats are expected to shrink throughout their range (Aryal et al., 2016; Li et al., 2016).

Mitigating human-snow leopard conflict through community engagement is one of the major objectives of the snow leopard conservation action plan for Nepal (2017–2021) (DNPWC, 2017). Research has shown that visitors are willing to pay for snow leopard conservation in the Annapurna Conservation Area in Nepal (Schutgens et al., 2018) but more research is needed on snow leopard interactions with human activities to better understand the influence of snow leopards on livestock herding practices and vice-versa (Alexander et al., 2016b).

This study was conducted to identify the major factors affecting the risk of livestock depredation from snow leopards. We also identified the potential snow leopard attack risk zone within the study area. We hypothesized that anthropogenic variables are correlated with livestock depredation risk from snow leopards than environmental and topographic factors.

Materials and Methods

Study area

The study was conducted in the southeastern part of Manang District, Nepal covering a total area of 235 km2 which is the jurisdiction of the District Forest Office (Now, Division Forest Office) (Fig. 1). We chose extent of study area by making three km buffering from livestock sheds. According to the herders and livestock owners, livestock travel for grazing up to three km and some of them travel in valley, rocks and glacier too. This distance was also validated and verified from district forest office staffs who regularly patrol there. Further, the snowfields around the attack zone is not permanent which allows seasonal grazing of livestock. Alongside, in rocky areas, small livestock like goat and sheep roam easily, thus considered in the buffer. The study area is rich in faunal and floral diversity. During the study, we recorded Himalayan pine (Pinus wallichiana), east Himalayan fir (Abies spectabilis), Himalayan birch (Betula utilis), yew (Taxus baccata), figwort (Picrorhiza scrophulariiflora), marsh orchid (Dactylorhiza hatagirea), caterpillar fungus (Ophiocordyceps sinensis), felworts (Swertia chirata), love apple (Paris polyphylla), sunpati (Rhododendron anthopogon), sea buckthorn (Hippophae spp.), lokta (Daphne bholua), lily (Lilium nepalense), black juniper (Juniperus indica) as the major plant species in the study area. Snow leopard (Panthera uncia), musk deer (Moschus moschiferus), common leopard (Panthera pardus), impeyan pheasant (Lophophorus impejanus), Himalayan goral (Naemorhedus goral), wolf (Canis lupus), Asiatic black bear (Ursus thibetanus), barking deer (Muntiacus muntjac), gray langur (Semnopithecus schistaceus) are the major wild animals found in the study area.

Figure 1 Study area.

Plot Design and Data Collection

Firstly, we visited all possible risk zones1 for the livestock depredation by snow leopard in the study area between April and June, 2018. We prepared the list of herders in the study area, and then conducted a workshop2 of 5–8 herders and 3–5 villagers3 to gather information related to livestock depredation by snow leopard and finally mapped the attack sites. Workshop with herders were conducted at livestock sheds and workshops with villagers were conducted at villages. These sites were further validated by district forest office (now division forest office) staffs to check the biasness, if any. The study area is the habitat of common leopard and Asiatic black bear as well; however, they use lower elevation than snow leopard and there is no habitat overlap4 in livestock attack zone. This was further confirmed by local herders, villagers and the forest staffs who regularly patrol there. Wolf generally hunt on pack (with group). Due to hunting patterns, information provided by herders and villagers, and verification by forest staffs, we confirmed that we collected locations attacked by snow leopard, not by other carnivores. A total of 109 attack sites in the last 3 years (2015–2018) were visited to record geo location (GPS points of all unique sites were taken). We maintained at least a 100 m distance between attack locations to ensure that each attack location is unique, resulting in 86 unique locations out of 109 collected.

Environmental Variables

Topographical variables

Geographic factors are responsible for the spatial distribution of the snow leopard (Wolf & Ale, 2009). These geographic variables were used to model the habitat of this species and other Himalayan carnivores in Nepal (Aryal et al., 2016; Bista, Panthi & Weiskopf, 2018; Panthi, 2018). A Digital Elevation Model (DEM) with 30 m resolution was downloaded from the United States Geological Survey (USGS) (https://earthexplorer.usgs.gov/). Slope and aspect were calculated from the DEM using ArcGIS software (Table 1).

Table 1 Environmental variables used for modeling.

Source	Category	Variable	Abbreviation	Unit	
USGS	Topographic	Elevation	elevation	m	
	Aspect	aspect	Degree	
	Slope	slope	Degree	
GEOFABRIK		Distance to water	dist_water	m	
MODIS	Vegetation-related	Mean EVI	evimean	Dimensionless	
	Maximum EVI	evimax	Dimensionless	
	Minimum EVI	evimin	Dimensionless	
	Standard deviation of EVI	evisd	Dimensionless	
GFC		Forest	forest	Dimensionless	
GEOFABRIK	Anthropogenic	Distance to livestock shelter	dist_goth	m	
		Distance to motor road	dist_motor	m	
		Distance to path	dist_path	m	
ICIMOD		Land use/land cover	landcover	m	

Vegetative variables

As the snow leopards are carnivores, their diet primarily consists of wild and domesticated herbivores (Aryal et al., 2014; Wegge, Shrestha & Flagstad, 2012; Weiskopf, Kachel & McCarthy, 2016), making vegetative variables important to consider (Andersen et al., 2000). Therefore, forest cover of Global Forest Change (GFC) (http://earthenginepartners.appspot.com/science-2013-global-forest) was used as a vegetative variable (Hansen et al., 2013). We also included Enhanced Vegetation Index (EVI) to model the potential attack risk of snow leopards. We downloaded EVI time series images from 2015, 2016, and 2017 from the Moderate Resolution Imaging Spectroradiometer (MODIS) sensor from the USGS. Then, we used Environment for Visualizing Images (ENVI) software to smooth the data by using an adaptive Savitzky-Golay filter in TIMESAT (Jönsson & Eklundh, 2004), which reduced the cloud effect and allowed us to obtain mean, maximum, minimum and standard deviation of EVI.

Anthropogenic variables

Large numbers of livestock are killed in Nepal due to the proximity of human settlements to the natural range of snow leopards (Aryal et al., 2014; Wegge, Shrestha & Flagstad, 2012). Assessing anthropogenic factors leading to livestock predation by snow leopards is critical as these are the variables that represent the greatest degree of control from humans and would allow for achieving the stated goals of snow leopard conservation and decreased livestock mortality from depredation. During field data collection, human activities were documented in snow leopard habitat. We obtained the shape file of motor roads and paths inside the study area from Geofabrik (https://www.geofabrik.de/data/shapefiles.html). The locations of livestock shelter within snow leopard habitat were collected during field work. Distance raster files of livestock shelters, footpaths, and motor roads were created using ArcGIS. We downloaded land cover and land use from the International Centre for Integrated Mountain Development (Uddin et al., 2015) and included them in the model.

Modeling livestock depredation risk from snow leopards

We used MaxEnt software to model the livestock depredation risk from snow leopard in the study area. Geo-referenced presence points of livestock attacks by snow leopard and the environmental variables (Table 1) were used as input variables to the MaxEnt model to produce a predictive livestock depredation risk map (Elith et al., 2006; Phillips et al., 2017; Phillips, Anderson & Schapire, 2006). The model was validated by the area under receiver-operator curve (AUC) (Pearce & Ferrier, 2000) and evaluated by True Skill Statistics (TSS) (Allouche, Tsoar & Kadmon, 2006). The multicollinearity between variables was less than 0.7, which is acceptable for modeling (Dormann et al., 2013). Seventy percent of the data were used to train the model and 30% were used to validate the model. We used 10 replications, 1,000 maximum iterations, and 1,000 background points during the modeling using reference from Barbet-Massin et al. (2012). The threshold to maximize the sum of specificity and sensitivity was used to calculate TSS and to prepare the binary map from the continuous map (Liu, White & Newell, 2013).

Results

Snow leopard attack risk zone

We identified 13.64 km2 as the potential risk zone for snow leopard attacks in the study area. The areas with the highest risk for snow leopard attacks were identified spatially using ArcGIS (Fig. 2). The AUC and TSS of the model were 0.941+/−0.013 and 0.862+/−0.047, respectively. A threshold of 0.273 was used to prepare the risk map from the continuous probability map.

Figure 2 Mapping of attack risk zone by snow leopard.

The most important variables found in the model to determine snow leopard attack risks are distance to livestock sheds, distance to path, aspect, and distance to motor road (Fig. 3). Other variables have less information to model the snow leopard attack risk. In Fig. 3, the regularized gain of the model without distance to livestock shed was less than that of the model using other single variables, therefore this is a more useful variable to the model. Similarly, the regularized gain of the models without distance to path, aspect and distance to road are less, which also demonstrates the high utility of these variables in modeling the snow leopard attack risk zone.

Figure 3 Importance of variables to train the model.

The regularized training gain explains how much better the model distribution fits the presence data relative to a uniform distribution. “With all variables” indicates the results of the model when all variables.

The regularized training gain of this figure explains how better the model distribution fits the presence data compared to a uniform distribution. “With all variables” indicates the outcomes of the model when all variables are used; “with only variable” denotes the effect of removing that single variable. “Without variable” denotes the result when only that variable is used (Phillips, 2017). See Table 1 for full variable names and descriptions.

Figure 4 Variable relationship and importance.

Snow leopard attacks on livestock are much more likely at closer distances to the shed which is used to house the livestock (Fig. 4A). Livestock are prone to attack by snow leopards far from foot paths and at a moderate distance to motor roads (Figs. 4B, 4D). At western aspect, the probability of snow leopard attacks on livestock is high in comparison to other aspect (Fig. 4C). Our study area was small; the relationships may be different in other regions or in case of a larger study area.

Discussion

The southern and western sections of the study area are at the greatest risk of livestock depredation from snow leopards. A portion of the west side of the study area is situated inside the Annapurna Conservation Area (ACA) and other patches identified as risk zones are very near to the ACA, which likely indicates that snow leopards living in the ACA may come to these places to prey on livestock. We did not assess the proportion of the habitat used by snow leopard inside and outside the protected areas, however this study somehow supports the finding of Deguignet et al. (2014). Their study depicted that small proportion (14–19%) of the species ranges in protected areas and primarily share the landscape with livestock herders. These patches were also identified by the interviewed respondents too as the good areas for livestock grazing, which is further evidenced by the high occurrence of livestock shelters. We surveyed only 235 km2, since the home range of this species is 124–207 km2 (Johansson et al., 2016), the total area may have retained only 2–3 individuals. However, a study indicates that Manang has 5–7 animals per 100 km2 (DNPWC, 2017), thus there might be more than 2–3 individuals in the study area.

Anthropogenic variables were identified as the most important factors influencing snow leopard attacks on livestock, and this finding is concurrent with existing literature. Multiple studies throughout the snow leopard’s native range have recorded livestock depredation (Li et al., 2013; Mijiddorj, Alexander & Samelius, 2018; Suryawanshi et al., 2013; Ud Din et al., 2017; Wegge, Shrestha & Flagstad, 2012) and, in Nepal, the spatial distribution of snow leopard activities has been positively correlated with human activities (Wolf & Ale, 2009). Additionally, carnivore food requirement and spatial needs often conflict with human interest which is the major challenge for biodiversity conservation and maintaining the viable population (Treves & Karanth, 2003), this is supported by this study too. Previous research has shown that the presence of livestock does not negatively affect the occurrence of snow leopards. Alongside, presence of higher livestock in grazing areas may have affected the space used by wild prey which could have forced the snow leopard to prey on livestock, which is supported by Karimov, Kachel & Hackländer (2018). In fact, snow leopards continue to hunt in the areas close to livestock herding (Alexander et al., 2016b; Rovero et al., 2018). The same finding was supported by the results of this study. Alongside, Johansson et al. (2015); identified that snow leopard preys on livestock mainly on stragglers and rugged areas where herders can’t pay attention for livestock.

Livestock predation by snow leopard is increased with livestock density (Suryawanshi et al., 2017). Similarly, our research shows that proximity to livestock shelters is the variable most closely associated with livestock depredation from snow leopards, which emphasizes the serious nature of human-snow leopard conflict. If snow leopards attack livestock far from livestock shelters, wildlife managers can restrict livestock from high risk areas and confine them to areas of relative safety using shelter. However, our findings indicate that the areas near to livestock shelter are at high risk of attack. Therefore, wildlife managers have to manage in such a way to allow for the co-existence of livestock and snow leopards in pastureland.

Our study also identified a higher likelihood of snow leopard attack far from foot trails and at a moderate distance from and motor roads. In the study area, there are many foot trails and a few motor roads. Generally, the foot trails and motor roads have a steady flow of traffic, resulting in few snow leopard attacks within direct proximity of foot trails and motor roads.

In order to achieve the goals of increasing conservation for snow leopards as well as decreasing livestock depredation and economic loss from snow leopard attacks, the findings from this research should be applied by herders in high risk zones. Given the high occurrence of attacks close to livestock shelters in the risk model, it is imperative that livestock herders utilize leopard-proof sheds and that vigilant care is given, even near the shelters. The data indicates that leopards are deterred from attacks in close proximity to humans, as shown by the lack of fit in the model at close spatial scales to footpaths and motor roads, demonstrate the efficacy of human presence in deterring snow leopard attacks. Further research is needed to determine if these results are applicable in areas beyond that studied in this investigation as well as identifying other factors and tactics that decrease human-snow leopard conflict.

It is noteworthy that MaxEnt software only considers the presence of livestock depredation by snow leopards and is therefore limited because the risk model cannot account for the absence of livestock attacks. Additionally, although we maintained at least 100 m in distance between livestock attacks to ensure their uniqueness, we did not fully avoid spatial autocorrelation. The statistical method to identify the minimum distance to deal with spatial autocorrelation may be useful for a more robust model. We have collected presence locations of attack sites based mainly on information provided by the herders and villagers, it may be biased and influence the result. Therefore, the probability of showing attack sites of their proximity may be higher. Furthermore, the snow leopard attack risk zone is also zone of livestock presence and livestock are vulnerable to snow leopard depredation in these zones. Finally, while the distribution range of snow leopards is extensive in the Manang district of Nepal, our study area is small and only represented a small portion of the large and heterogeneous district.

Conclusions

This study identified the risk zone of snow leopard attacks in the Manang district of Nepal. The southwestern part of the study area was found as the most vulnerable to snow leopard attacks. The distance to livestock shelters, distance to paths, aspect, and distance to roads are the most important variables in defining the risk of snow leopard attacks in the study area. Snow leopards prefer to attack livestock near livestock shelters and at moderate distances from roads. These identified risk patches should be managed to conserve both the snow leopard and to protect the livestock. The herders should be encouraged to protect their livestock through active caretaking, even in close proximity to livestock sheds, and keeping them in leopard proof sheds, which will result in less human-snow leopard conflict. Investigations of this nature should be conducted throughout the snow leopard’s range to determine the factors affecting livestock depredation by snow leopards and to model snow leopard attack risk zones across its native range.

Supplemental Information

Supplemental Information 1 Dataset of snow leopard attack sites

Detailed data on attack sites, livestock shed and floral and faunal presence in the study area.

Click here for additional data file.

We would like to thank all the staff of the District Forest office in Manang for their support during the entire study, especially during data collection, as well as the communities that participated in the study. We would like to thank Marcus E. Taylor, Rajani Regmi and Shambhu Paudel for reviewing the language of the manuscript.

Additional Information and Declarations

Competing Interests

Author Contributions

Human Ethics

Animal Ethics

Data Availability

1 Possible risk zones were identified based on proximity to shed, grazing/browsing pastures of livestock, grazing area where herders are normally absent.

2 A total of 17 workshops were conducted and participants were replicated in some workshops.

3 Only villagers who received the compensation for livestock depredation were considered for workshops to make sure that participants know the real information about attack sites.

4 Most of the attack zones are around 4,000 m of altitude.

The authors declare there are no competing interests.

Ajay Karki conceived and designed the experiments, performed the experiments, analyzed the data, prepared figures and/or tables, authored or reviewed drafts of the paper, and approved the final draft.

Saroj Panthi analyzed the data, prepared figures and/or tables, and approved the final draft.

The following information was supplied relating to ethical approvals (i.e., approving body and any reference numbers):

Department of Forests, Nepal. District Forest Office, Manang.

The following information was supplied relating to ethical approvals (i.e., approving body and any reference numbers):

There was no direct animal involvement in this study, however this program/project was approved by the Government of Nepal, Department of Forest.

The following information was supplied regarding data availability:

The raw data is available in the Supplementary File.

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
