# Peer review of "Factors affecting livestock depredation by snow leopards (Panthera uncia) in the Himalayan region of Nepal"

_PeerJ, doi:10.7717/peerj.11575_

## Round 0.1 · original submission · Major Revisions

I hesitated between rejecting your manuscript and major revision as there were major issues in your paper, in terms of lack of description of your design and methods (remember that one should be able to replicate your study if needed), as well as interpretations of the results - there is nothing wrong with alternative interpretations of the same patterns, as long as you can show that your data do provide relevant information (for example that do not suffer from large selection bias that could not be taken into account). The reviewers have spent time reading and making suggestions for how to improve the paper - you absolutely need to provide a convincing answer and changes if you want your paper to be published in PeerJ.

·

Basic reporting

With a few important exceptions, detailed in my General Comments, the authors employ clear and unambiguous English. Where this is not the case, easy changes are possible (for example, replace “group works” with “workshops”), which I have noted elsewhere.

Experimental design

Methods are not described with sufficient detail to replicate.
Additional details of the study design are needed for the reader to understand if and how you have adequately accounted for unavoidably opportunistic sampling approach.
There are multiple steps in the sampling process with the potential to introduce substantial bias and influence the results, but which do not appear to have been recognized.

Validity of the findings

You have not reported the key data underlying your manuscript - the locations of livestock attacks reported. Instead you have provided auxiliary data on the locations of livestock shelters, and of plant and wildlife sightings or sign - data which are not used in the manuscript. While these are helpful, they are inadequate on their own - you report collecting 109 conflict locations, which you truncated to 86 “unique locations.” These data should be provided for review, and as a map in the manuscript. Additionally other information is needed to assess and interpret the validity of assumptions made implicitly in your MaxEnt modelling approach. See my general comments for more details.

Additional comments

SUMMARY
The authors use MaxEnt presence-only modelling to explore potential relationships between anthropogenic, biotic, and physical spatial covariates and snow leopard attacks on livestock in Central Nepal. They collected precise location data for over 100 putative (but unconfirmed, and unreported) snow leopard attacks, from which they infer that snow leopards prefer to attack livestock near barns, trails and roads. This finding has potentially important implications for conservation efforts aimed at reducing such interactions, but further details are needed to convince future readers that these findings are representative of the underlying ecology rather than sampling effort.

While the data set is commendable, and the authors are mostly restrained in their interpretations (In particular I appreciate the circumspection offered in Lines 149-156), the analysis and subsequent interpretation should be reconsidered in light of some unavoidable limitations and confounding factors that have not been adequately addressed. Foremost among my concerns is the possibility that the data and findings reflect sampling effort, not biological phenomena. The two dominant themes underlying my concerns are availability (Johnson et al. 1980), and imperfect detection (Mackenzie et al. 2002).

While I cannot recommend the manuscript for publication in its present form, I see opportunity for revisions that could better reflect uncertainties that are inherent in the data, and thus provide more circumspect and interpretable insight into the phenomena of snow leopard depredation on domestic livestock, a problem that has proven to be a major driver of anthropogenic mortality for the species.

MAJOR CONCERNS

I. Raw attack location data are not provided.

II. Consider your central finding that snow leopards “prefer to attack livestock near livestock shelters…” Unfortunately, there are much simpler, more parsimonious explanations for the patterns in your data that have not been accounted for in your analysis. (To be clear, I think the following possibilities need to be addressed and ruled out if you want to convince the reader of your interpretation.)

1. The recorded locations of snow leopard attacks reflect where you spoke with herders and villagers. In other words, although you are trying to map the risk of snow leopard attacks on livestock, you have instead mapped the distribution of those locations where you sampled (or more likely, some combination of sampling effort and attacks). As explained in more detail by Elith et al. (2011):

“A second fundamental limitation of presence-only data is that sample selection bias (whereby some areas in the landscape are sampled more intensively than others) has a much stronger effect on presence-only models than on presence-absence models...This bias will most commonly occur in geographic space (e.g., close to roads) but could be environmentally based (e.g., visiting wet gullies) but, regardless, will map into covariate space. Under biased sampling, a presence-only model gives an estimate of [presence X sampling] rather than [presence]. That is, we get a model that combines the species distribution with the distribution of sampling effort...”

Currently, the manuscript is missing important details of the sampling approach (and the raw data) which are needed to help the reader rule out the explanation above. You state that you assessed a total of 454 km2 yet no details are provided that demonstrate how the area was sampled. Indeed, you do not even provide a map of the attack location data. Given the structure of the terrain (2000-8000 m elevation!), and the nature of the data collection, no one would expect random or even sampling of the landscape. But even though you spoke to people to get your “samples,” you are still, ultimately, sampling the landscape, and should therefore show your reader how that sampling was distributed (. Because the sampling process involves 2 steps (A. a herder/villager observes a snow leopard attack, and B. You talk to that person and they report the attack), I think you will need to provide hopefully 2 pieces of information (as a map preferably) to convince the reader: A. Where on the landscape do the people you spoke to go with their livestock, or, where do livestock graze [in other words, where are the villagers/herders sampling? This information is only partially provided in the supplemental file with livestock shed locations],and B. Where did you talk to villagers/herders [where did you conduct the workshops?].

2. Alternatively (or additionally), the recorded attack locations simply reflect livestock distribution on the landscape. This is related to the detectability issue in the previous possible explanation, but also related to the concept of availability. To illustrate, a snow leopard might prefer to kill livestock far from livestock shelters and people, but, because there are no livestock in those places, you do not have any records of snow leopard attacks there. A reader could interpret your current presentation to mean that there would be fewer attacks on livestock if people simply moved their livestock farther away from shelters, roads, and trails. This clearly problematic conclusion follows logically from your current presentation. ( A more logical conclusion might be that people should move the location of shelters, roads and trails - but I’m sure you can see why that is problematic too).

I think this is also related to the “possible risk zones” mentioned on line 88. In a footnote, you describe that these "were identified based on proximity to shed, grazing/browsing pastures of livestock, grazing
area where herders are normally absent." A map of those would be helpful to better understand the implication of this filtering process, and such a map probably represents the actual extent of your study area (see below), for the purposes of MaxEnt modelling.

While presence-only data can indeed be used to infer preference if detectability has been accounted for, such inference is well-understood to be sensitive to the resources deemed “available.” In your analysis, the “available” resources are represented by the landscape used in the MaxEnt model. But without more details about the sampling process and the study area, we have no way of knowing whether this landscape was truly available for snow leopards to kill livestock. Again, given the huge variation in elevation, it is very safe for the reader to assume that livestock are not present throughout the putative study area. Meaning that to address your stated goal of understanding what factors influence where snow leopards kill livestock, you will be better served by limiting the MaxEnt landscape to include only those parts of the landscape grazed/used by the villagers and herders you spoke with. In its present form, you are probably modelling where livestock occur, instead of where attacks occur (given livestock presence). Symbolically, I think you want to know what covariates influence the conditional probability P(attack|livestock presence) which is relevant to understanding where livestock are vulnerable to snow leopard depredation, rather than where in the world snow leopard depredation occurs (by defniition, it can only happen in places where there are snow leopards and livestock). What you may have modeled is where livestock are, P(livestock presence), in a landscape where snow leopards are presumed to be present. I think the best option is to define a more limited landscape for which you can assume that P(livestock presence)=1, and avoid any chance of confounding presence and attacks. Thus, I suggest that you rerun your MaxEnt models with a landscape that only includes areas where livestock are plausibly present in this landscape (I think this is the “possible risk zones” that you mention).


MINOR CONCERNS:
[I have not provided Line by line feedback, as major analytical revisions will require substantial revision of all sections of the paper]

1. If, after accounting for the distribution of livestock and sampling effort in the landscape, you still find that proximity to livestock shelters roads and trails are the most important factors affecting attack occurrence, what would account for that pattern? Why would snow leopards not attack livestock farther from human infrastructure? Alternatively, suppose that the pattern in your data has more to do with where livestock occur (and that because this critical piece is not included in your model, the other factors you’ve identified come up as the next best indicators). What would that indicate?

2. Figure 1 contains very little information, and somewhat misleadingly implies that the study area is a homogeneous landscape without any major structural variation. I suggest that the current figure 1 serve as an inset for a more detailed map of the study area showing some of the information outlined in my earlier comments, against a background that provides context to the structure of the terrain (e.g., a DEM, or aerial imagery)

3. You have not discussed Suryawanshi et al.’s (2017) paper, and its central finding that livestock predation increases with wild ungulate and livestock density. The scale of the analyses in the their paper is very different, but in light of their findings, I think it would be helpful if you could provide even general contextual information about the density of wild and domestic ungulates in your study area compared to other nearby areas, and consider how this might influence interpretation and general applicability of your results.

4. What (if any) criteria were used to distinguish common leopard, wolf, and bear attacks from snow leopard attacks? Presumably people did not always see the animal that attacked their livestock, or could have mistaken a snow leopard and common leopard?

5. Scale: The area of likely snow leopard habitat in the study area (i.e. excluding heavily glaciated and high elevation portions of the study area) is quite small (250km2) relative to a snow leopard home range (also roughly 200-250km2 based on Johansson’s estimates from Mongolia - Jackson’s VHF-based estimates cited for Nepal (cited on line 37) are regarded even by Jackson himself to be unreliable and an artifact of old technological limitations). This raises the question that the patterns in your data may not be very representative of the broader regional landscape. (Which isn’t necessarily a bad thing, but is a limitation that should be discussed), but rather individual variation among just a few snow leopard individuals (as you rightly note the population in your study area may only be 2-3 animals)

6. 109 unique attacks in a 3 year period suggest that a major proportion of snow leopard diet in the study area is made of livestock. Do you think this may impact the spatial depredation risk you are modeling in important ways?

7. “Group works” is a confusing term to me (e.g., Line 90). I suggest replacing with “group interviews” or “workshops”

8. Line 209-210 is not clear.

9. Footnote 3: “only villagers who received compensation….” what, if any, additional bias might this have introduced to your data? Is there any kind of verification involved in certifying that a snow leopard killed the livestock before compensation is paid? Is there a bias in that process?

10. Figures 3 and 4: Change variable names and axis titles to make them more understandable (e.g. Distance to footpath instead of dist_path)

Some additional references to consider -
Elith, J., Phillips, S.J., Hastie, T., Dudík, M., Chee, Y.E., Yates, C.J., 2011. A statistical explanation of MaxEnt for ecologists: Statistical explanation of MaxEnt. Diversity and Distributions 17, 43–57. https://doi.org/10.1111/j.1472-4642.2010.00725.x

Suryawanshi, K.R., Redpath, S.M., Bhatnagar, Y.V., Ramakrishnan, U., Chaturvedi, V., Smout, S.C., Mishra, C., 2017. Impact of wild prey availability on livestock predation by snow leopards. Royal Society Open Science 4, 170026. https://doi.org/10.1098/rsos.170026

Reviewer 2 ·

Basic reporting

Technically the chosen study area for snow leopard conflict study is not correct. The manuscript lacks some of the recent literature and updates on the topics in this field. The given illustrations are very limited and difficult to understand. Also, data given in the supplemental files is very limited and difficult to link with the manuscript.

Experimental design

It was not clear how the data has been collected? It says 17 group discussion was done. The participants involved in the group discussion belong to which area/ villages or municipalities. The study area is not defined properly. The study area needs to be described clearly with an appropriate figure (s). Which area/landscape the authors is talking about is very important to make it clear to the audiences. Also, the variables used in the models are not enough to conclude with a big statement. Prey is one of the most important variables which is missing in the model. This has to be incorporated in the model to draw a meaning conclusion, without incorporating prey density/abundance, the study is incomplete.

Validity of the findings

It was a good idea to known the risk zone and high attack sites but the variables used are incomplete. Also, it was not clear how the participants were selected from the study area and how the data has been validated in the field.

Additional comments

It was a good study given that the choice of the study area is appropriate. It was very difficult to understand where the sampling was done and who are the participants of the group discussion (villages/municipalities). The most important part of the methods was not clear i.e. selection of the participants for discussion and mapping conflict areas. The authors need to define the study area very clearly. Also, it will be good to mention why this area was selected for this study. The variables used in the model is incomplete, the authors should include prey abundance/density as one of the variables in the model. Without this, the model is incomplete. I suggest the authors re-run the models incorporating this variable to draw a meaningful conclusion.

---

## Round 0.2 · Major Revisions

The reviewer acknowledges the effort you put in revising the paper but it does not answer his/her main concern regarding potential biases in the way you use and interpret the model. The reviewer provides two clear solutions in terms of revision, and I strongly urge you to argue for choosing one or the other and change the manuscript accordingly.

·

Basic reporting

no comment

Experimental design

See my general comments for further thoughts on this. As before, the sampling process has the potential to introduce substantial bias and influence the results, and have not been fully considered.

Validity of the findings

There is a mismatch between your interpretations and the extent of the background area used in your MaxEnt Model. To resolve this in my opinion will require either A. rebuilding the model with a more carefully defined extent, over which you can more confidently assume even sampling probablility, or B. revising the manuscript to reflect that there were likely major sampling biases relative to the extent used, biases which limit and confound your interpretations.

Additional comments

The additional details provided are very helpful and alleviate many of the concerns that I raised in my first review.

However, in my first review I highlighted my concerns about potentially important sampling bias confounding your results and I suggested that you to build a new model with a more limited geographic extent as one possible solution to resolve these issues. I will limit my remarks to this topic, as I remain unconvinced that you have addressed all my concerns.

I see from your rebuttal letter that there is some confusion about what I meant by "sampling." If you are mapping a phenomena, then the “population” that you are sampling from is the population of possible locations on the landscape. In other words – although you are speaking with human beings, you are sampling the landscape (very specifically in the context of MaxEnt models, you are sampling raster grid cells where snow leopard attacks occurred).

The landscape - geographic extent of your model - is equally as important as the data you feed into the model. The extent that you use (and the grain or resolution), can have huge influence on your conclusions, and as such, you should justify your choices, and recognize how those choices impact your conclusions and interpretations. Presently, the manuscript lacks any justification for the chosen extent, and does not recognize the limitations that the chosen extent imposes on the interpretations.

In your introduction you state the purpose of the study was to identify the factors affecting the risk of livestock depredation by snow leopards. Elsewhere, you interpret your findings as indicative of where snow leopards prefer to attack livestock. Together, this suggests to me that you are interested in understanding livestock depredation locations in comparison to all locations used by livestock – thus I suggest that you limit the extent to those areas (even if you have to make some assumptions and use a fixed buffer around livestock shelters or something).

Your model assumes that you had equal opportunity to observe livestock depredation at every point within the model extent. The extent of your Maxent model is presently constructed in such a way that you are almost guaranteed to find “that anthropogenic variables are more highly correlated with livestock depredation risk from snow leopards than environmental and topographic factors,” your stated hypothesis. This is because your model presently confounds snow leopard attack probability with livestock presence probability. This confounding arises out of sampling bias and an overly broad extent, which may be due in part to confusion about what you sampled.

For your results to truly reflect where livestock are vulnerable to snow leopard attack, you need to account for where livestock were present. You identify distance to livestock shelters, roads, and paths, as well as elevation, as the most important variables influencing snow leopard risk to livestock. Unfortunately, the more parsimonious explanation is that this is not snow leopard risk, but instead where livestock were present, i.e., where you sampled the landscape. In other words, the extent of your MaxEnt model is not matched to the extent of the area from which samples were drawn.

I see two options:
1. Change the model. This is the best option in my opinion. As I said in my initial recommendations, I still think you should run the model again, using a much smaller landscape that excludes places where livestock are not present, for example, by only considering areas that fall within a set buffer distance of livestock shelters.
2. Alternatively, if you are willing to shift your interpretation further, and avoid talk of preference or risk, you could change the manuscript to reflect that you modeled where snow leopards killed livestock, but, because you did not account for where livestock are present in the study area you can only speculate as to why that was the case. This approach will require changes throughout the manuscript. For example, If you are comfortable with this confounding, you should change statements such as “Snow leopards prefer to attack livestock near livestock shelters and at moderate distances from human paths and motor roads”, to instead say something more like, “snow leopard attacks on livestock occurred near livestock shelters, but because we did not account for livestock availability, we were unable to determine if this pattern reflected an intrinsic preference on the part of snow leopards, or simply where livestock were most likely to be found.” The management implications of these two things are very different. The preference hypothesis implies that moving livestock to higher, further pastures would reduce attacks. The livestock presence hypothesis implies that livestock need better protection everywhere. Without accounting for livestock presence in defining the model’s geographic extent, you can only speculate - you do not have quantitative ability to distinguish between these two. Intuitively, the livestock presence hypothesis is more believable. That’s why you should probably just change the model. It will make for a much better manuscript. But a more restrained and circumspect presentation that recognizes the limitations of the model as it currently stands is nonetheless possible.

---

## Round 0.3 · Minor Revisions

Thanks for making some of the changes required by the reviewer, but they were minimalistic. You need to better justify the buffer used now (3 km rather than 5), and not just make the change without further comment. Looking at your figures and the valleys/glaciers still included in the buffer, I am not convinced you made the changes asked for by the reviewer.

You need also to change figure 4 so as to remove negative values for distances or aspect (and then removing the red lines that go through 0). You need also to comment that the shape of these relationships is most likely an artefact of the region used to fit MaxEnt, that is they provide some rough guide but not the precise relationships as seems to be indicated by the blue ribbons (what are they?).

---

## Round 0.4 · Minor Revisions

The abstract and the text require some minor revisions. I am not convinced by your justification of the 3 km buffer - now you claim that livestock move up to 3 km (without providing any evidence for that), but then why did you use 5 km in the previous version?

---

## Round 0.5 · accepted · Accept

I appreciate your patience in revising the manuscript multiple times, but it was important to clarify some points.